# Enterprise Architecture as Explanatory Information Systems Theory for Understanding Small- and Medium-Sized Enterprise Growth

**Aurona Gerber** [1,2,*], **Pierre le Roux** [1,3] **and Alta van der Merwe** [1]

1   Department of Informatics, University of Pretoria, 0083 Pretoria, South Africa;
    pierre.leroux@moyoafrica.com (P.R.); Alta@up.ac.za (A.v.d.M.)
2   CAIR, Center for Artificial Intelligence Research, 0083 Pretoria, South Africa
3   Moyo, 0157 Centurion, South Africa
*   Correspondence: aurona.gerber@up.ac.za

**Abstract:** Understanding and explaining small- and medium-sized enterprise (SME) growth is important for sustainability from multiple perspectives. Research indicates that SMEs comprise more than 80% of most economies, and their cumulative impact on sustainability considerations is far from trivial. In addition, for sustainability concerns to be prioritized, an SME has to be successful over time. In most developing countries, SMEs play a major role in solving socio-economic challenges. SMEs are an active research topic within the information systems (IS) discipline, often within the enterprise architecture (EA) domain. EA fundamentally adopts a systems perspective to describe the essential elements of a socio-technical organization and their relationships to each other and to the environment in order to understand complexity and manage change. However, despite rapid adoption originally, EA research and practice often fails to deliver on expectations. In some circles, EA became synonymous with projects that are over-budget, over-time and costly without the expected return on investment. In this paper, we argue that EA remains indispensable for understanding and explaining enterprises and that we fundamentally need to revisit some of the applications of EA. We, therefore, executed a research study in two parts. In the first part, we applied IS theory perspectives and adopted the taxonomy and structural components of theory to argue that EA, as represented by the Zachman Framework for Enterprise Architecture (ZFEA), could be adopted as an explanatory IS theory. In the second part of the study, we subsequently analysed multiple case studies from this theoretical basis to investigate whether distinguishable focus patterns could be detected during SME growth. The final results provide evidence that EA, represented through an appropriate framework like the ZFEA, could serve as an explanatory theory for SMEs during start-up, growth and transformation. We identified focus patterns and from these results, it should be possible to understand and explain how SMEs grow. Positioning the ZFEA as explanatory IS theory provides insight into the role and purpose of the ZFEA (and by extension EA), and could assist researchers and practitioners with mediating the challenges experienced by SMEs, and, by extension, enhance sustainable development.

**Keywords:** enterprise architecture; information systems theory; theory of explanation; SME growth and transformation; Zachman framework for enterprise architecture

## 1. Introduction

It is a sobering fact that approximately 90% of all small businesses fail [1–3]. Furthermore, at least 80% of business in most economies are considered small or medium-sized enterprises (SMEs) [4].

Most large businesses started out as SMEs in some way, and due to their role in an economy, SMEs are increasingly recognised as contributors towards sustainability because their cumulative impact is far from trivial [5–7]. In developing countries such as South Africa, SMEs play a major role to solve socio-economic challenges, such as an unemployment rate of more than 28% [8]. Research also indicates a direct relationship between SME success and their prioritisation of sustainability considerations [6]. Within this context, "sustainability" refers to the definition by Klewitz and Hansen as "the configuration of business strategies and practices that contribute to sustainable development by endorsing social cohesion and environmental conservation in the long-term while simultaneously meeting the economic imperatives of profitability and growth" [5]. Understanding and explaining SME growth and success are, therefore, important topics for scientific research including the nature and management of SMEs, as well as the related sustainability impact. To execute such scientific research studies, it is necessary to, in the first place, consider relevant theory, and in the second place, the domains and existing body of knowledge on enterprises where an enterprise is defined as any socio-technical organization [9,10].

Scientific research is grounded in theory, and discussions about what is meant by theory and how theory underpins scientific knowledge are present in all disciplines. The information systems (IS) discipline studies socio-technical phenomena and the role of theory and the nature of theory in the IS discipline are popular discussion topics within recognised IS publications [11–20]. This is in part because IS is the discipline that is at the intersection of the knowledge about physical objects (machines) and the knowledge about human behaviour [19], and is, therefore, influenced by theories from different foundational disciplines. The longstanding discussions on theory culminated in publications by Gregor about the structure and nature of theory within IS [19,21], and this perspective has been adopted within this study. According to Gregor, some of the primary functions of theory include to describe phenomena of interest, analyse relationships between constructs in order to understand and explain [19] and such theories are therefore particularly suitable for understanding SMEs.

When one considers the relevant fields that are pertinent to SMEs, enterprise architecture (EA) emerges as one of the dominant research domains. Since its origins in the late 1980s, EA developed as a comprehensive field receiving significant interest from scholars and practitioners. Today, the EA domain is characterized by a plethora of publications and discussions, frameworks, methodologies and practices. The definitions of exactly what EA entails evolved with the field, but nowadays most of the accepted definitions include notions such as that EA is the continuous practice of describing the essential elements of a socio-technical organization and their relationships to each other and to the environment, in order to understand complexity and manage change [22,23]. This definition of EA is adopted for this study. Nevertheless, most publications and discussions on EA are concerned with the development and implementation of EA practices and processes [23–29], and not with understanding and explaining. EA is a wide field with many different frameworks and perspectives. However, if we focus on the architecture of enterprise architecture, the departure point of EA is the descriptive representation of an enterprise, and the Zachman Framework for Enterprise Architecture (ZFEA) is arguably the foundational basis or ontology focusing primarily on the nature and purpose of descriptive representations—for this paper, we adopt the ZFEA. If we, furthermore, aim to remain true to the fundamental intent of EA, and one considers the functions of IS theory that describes and explains towards understanding, our research questions emerge, namely:

- Can EA, when represented through an appropriate framework such as the ZFEA, be considered an explanatory IS theory?
- If it is the case that the ZFEA may be adopted as an explanatory theory, is it possible to describe and understand SMEs and subsequently explain SME growth?

In order to answer the above questions, we conducted a research study that consists of two parts that address the respective research questions. In the first place, we extended and summarised previous works that apply Gregor's structural analysis and taxonomy of theory to the ZFEA. The analysis found that the ZFEA conforms to the structure and function of an explanatory IS theory. In the second place,

we adopted the ZFEA as a theoretical basis for analysing successful SMEs' growth in multiple case studies. Across participating SMEs, the analysis detected focus patterns on ZFEA-related organisational elements and concepts during growth and transformation, thus, supporting the argument that EA as descriptive representation may serve as an explanatory theory for understanding SMEs and SME growth. This work is unique because EA itself has never been positioned as a theory. If EA as theory could assist with understanding and analysing, as well as explain outcomes of interventions, it could be an invaluable foundation for enterprises within the age of digital transformation

The remainder of this paper is structured as follows. In the literature review section, we provide relevant background on IS theory, EA, the ZFEA, and SMEs. In Section 3, we address the first research question and position the ZFEA as an explanatory theory. In Section 4, we present the research method, data collection and analysis for the second part of the study. This part addresses the second research question and analysed multiple SME case studies from the basis of EA represented by the ZFEA as an explanatory theory. In Section 5, we present the results and findings, followed by Section 6 where we discuss implications of the study. We conclude in Section 6.

## 2. Literature Review

In this section, we first provide a short introduction to perspectives in IS theory, in particular, that of Gregor that presented a taxonomy of the types of theories within IS with their distinguishing characteristics. The next section provides a description of EA followed by a summary of the ZFEA, in particular. The last section in the literature review provides relevant background on SMEs.

### 2.1. Information Systems Theory

We adopt an information systems (IS) perspective on theory, where IS is the academic discipline concerned with the study of socio-technical organizational systems that classically include four components: task, people, structure (or role), and technology [30]. IS as a socio-technical discipline emerged a few decades ago and was originally primarily concerned with data-centric systems (often mainframes) and their functions. Several definitions of IS still allude to these origins, with Encyclopaedia Britannica stating that an information system is "… an integrated set of components for collecting, storing, and processing data and for providing information, knowledge, and digital products" [31]. With the rapid development of computer technology and systems, IS evolved through several distinct eras with specific research interests as documented by Hirschheim and Klein [32]. What is notable in IS history is the influence of theory from relevant sister disciplines on IS theory. Whilst IS theory originally mostly resembled functionalist theory from the natural sciences [33], the development through management information systems (MIS) and organisational management incorporated theory from the social sciences [16,34], as well as sciences of the artificial and design sciences [35,36]. Nowadays, IS has a wider scope including the societal, organisational, and business impact of technology—including digital transformation [37]—and IS theories are used to analyse, predict, explain and/or prescribe [19,38,39].

Within IS, Gregor is specifically acknowledged for her work on the nature of IS theory regarding both theory types and theory components [19,38,40,41]. Within IS, the role of theory is to enhance our understanding of the world by providing explanations, descriptions, predictions and actionable guidance [19]. Gregor's work is foundational for most recent discussions about theory in IS [42–46] and she proposed a classification schema or taxonomy of IS theories based on core characteristics and the primary goal of the theory [19]. The primary goal of a theory is directly related to a question or a problem that needs to be investigated, and theories are, therefore, developed for the purposes of analysis and description, prediction, explanation and prescription. Table 1 provides the taxonomy of the IS theory types with the associated goals. IS theories, furthermore, have distinguishable components that make them theories (Table 2).

**Table 1.** A Taxonomy of Theory Types in Information Systems Research (Reproduced from Reference [19]).

| Theory Type | Distinguishing Attributes and Goal |
| --- | --- |
| I. Analysis | An *analysis theory* is a theory that states "what is" and focuses on analysis and description only. An analysis theory does not include predictions or indications of causal relationships among occurrences/events/objects. |
| II. Explanation | An *explanation theory* is a theory that states "what is", "how", "why", "when" and "where". The main aim is one of explanation and to provide understanding. An explanation theory provides explanations but does not aim to predict with any precision, and the theory is not testable. |
| III. Prediction | A *prediction theory* is a theory that states "what is" and "what will be". The theory provides predictions and has testable propositions but does not have well-developed justificatory causal explanations |
| IV. Explanation and prediction (EP) | An *explanation and prediction (EP) theory* states "what is", "how", "why", "when", "where" and "what will be". The theory provides predictions and has both testable propositions and causal explanations. |
| V. Design and action | A *design and action theory* is a theory that states "how to do something". The theory gives explicit prescriptions (e.g., methods, techniques, principles of form and function) for constructing an artefact or complex object. |

**Table 2.** Structural Components of Theory (Reproduced from Reference [19]).

| Theory Components (Common to All Theory Types) | Definition |
| --- | --- |
| Means of representation | A physical representation of theory. This might include mathematical terms, symbolic logic, tables, diagrams, graphs, illustrations, models, prototypes. |
| Constructs | The focus point or object of the theory. All primary constructs in the theory should be well defined. Many types of constructs are possible, e.g., observational (real) terms, theoretical (nominal) terms and collective terms. |
| Statements of relationship | The nature of the relationship among the constructs depends on the purpose of the theory. Types of relationships: associative, conditional, compositional, unidirectional, bidirectional or causal. |
| Scope | The scope is specified by the degree of generality of the statements of relationships and statements of boundaries showing the limits of generalisation |
| **Theory Components (Contingent on Theory Type)** | **Definition** |
| Causal explanations | The theory gives statements of relationships among occurrences/events/objects that show causal reasoning (not covering law or probabilistic reasoning alone). |
| Testable propositions (hypotheses) | The relationships between objects/events (constructs) can be tested through observation or experience. |
| Prescriptive statements | The theory provides a method or guidance on how to accomplish something in practice, e.g., construct a complex object or develop a strategy. |

Gregor also proposed a method to evaluate the proposed taxonomy by classifying five different accepted IS theories and used the component structure of Table 2 to analyse their structural components [19]. The method Gregor presented and adopted herself for theory analysis has been used

extensively since publication to classify and analyse theory within IS with regard to phenomena of interest [40,47–51]. To answer the first research question of this study, we adopted this method as used by Gregor to classify the ZFEA as an IS theory type (refer to Section 3).

*2.2. Enterprise Architecture*

Enterprise architecture (EA) as a research field emerged during the late 1980s when the adoption of information technology (IT) systems by organizations to support business functions became a significant concern. Different disciplines expressed the need to describe, understand, represent and design different dimensions of an organization. During this time, John Zachman (often called the father of EA) proposed the Zachman Framework for Enterprise Architecture (ZFEA) defined as a descriptive, holistic representation of an enterprise to provide insights and understanding where an enterprise is widely defined as any socio-technical organisation [9,10]. Since its origin, EA developed as a comprehensive discipline receiving significant interest from scholars and practitioners [24,52–55], and development included several EA frameworks, methodologies and practices [28,56–58,58]. EA has been incorporated into organizations from various perspectives and supporting different functions. These EA implementations included understanding an organization with its different components and their interrelationships, aligning the different business components (typically business–IT alignment), optimizing business functions, facilitating organisational integration, and enabling strategic management [10–16]. Recent developments in EA include enterprise architecture management (EAM) that elevates EA as a strategic business function assisting with organisational agility and transformation [24,59–63]. This transformation incorporates digital transformation since EA always had as a primary concern computer systems or information technology (IT) within organizations, and the role of EA for digital transformation has received particular attention [64–68]. EA frameworks often design the transformation of an organization moving from an "as-is" to a "to-be" state of design, with digitization and digitalization often drastically influencing various aspects of a business on its digital transformation journey. The impact of these influences and changes needs to be understood and designed to effectively maintain business functions, remain relevant and keep a competitive edge—which has always been a primary objective of EA.

However, EA is not without challenges. EA experienced a rapid original adoption typical of technological development but unavoidably slipped down the trough of disillusionment as indicated by the Gartner EA Hype Cycle [69]. Although many scholars asserted the need for EA to ensure successful business structures or business–IT alignment, many EA initiatives became synonymous with projects that are over-budget, over-time, costly and not delivering the expected outcomes [70–72]. A few decades of research still show different opinions regarding the effectiveness of EA to solve real-world challenges [73,74] and despite more than 2 million scholarly publications retrieved from Google Scholar (A search on Google Scholar executed on 22/7/2020 for "enterprise architecture" returned "About 2 320 000 results (0,07 sec)"), Gartner in 2018 depicted EA as slowly emerging from the trough of disillusionment [69]. Any potential EA adopter would also be confronted with a vast number of perspectives and the associated definitions of EA and EAM [22], as well as many and diverse EA frameworks, tools, methodologies and practices [39,75].

In this paper, we support the argument that EA remains indispensable for organisational transformation, including digital transformation, but that we fundamentally need to revisit the role of EA. We argue that, to mature and form a cohesive discipline that addresses the challenges in the age of digital transformation, EA as a discipline requires theory. This sentiment is supported in recent work by Radeke [39] who did a comprehensive literature analysis and found that EAM literature is dominated by prescriptive research and that "explanation and prediction is negligibly small". Discussions about EA theory are limited and generally study the impact of EA and EAM from organisational theory perspectives [76–79], discuss the systems theoretical basis of EA [80,81] or adopt a prescriptive design theoretical perspective for EA management practices [39,82,83] as is discussed in the next section.

*2.3. Enterprise Architecture and Theory*

There is a lack of research and associated literature that discuss EA and theory. In this section, we provide a summary of relevant perspectives on EA and theory, specifically IS theory since EA as a socio-technical phenomenon is mostly positioned and studied from an IS perspective.

- **EA for understanding enterprises**: Within this broad perspective, EA is used as a mechanism to understand organisations and subsequently propose interventions [26,57,84–92]. None of the literature that reflects this position adopts the position that EA is a theory even though the use of EA to understand and describe a phenomenon supports the notion of EA as a theory.
- **Using organizational theory to evaluate EA implementation and impact**: This perspective adopts the position that EA is an organisational intervention and research studies apply organisational theory to understand and evaluate the impact of different EA initiatives as well as prescribe management practices from a design theoretical perspective [30,65,76,77,83–85,93–96]. This perspective, therefore, focuses on the practical and procedural aspects of especially EA frameworks such as the Open Group Architecture Framework (TOGAF) [97].
- **The paradigm of EA**: Some of the original discussions about the nature of EA refer to the fundamental departure points and argue that EA adopts a systems theory paradigm [80,81]. This is supported by Zachman that originally developed the ZFEA from a systems engineering perspective, in other words, viewing the enterprise as a system consisting of interrelated components.
- **EA and theory**: Few publications discuss theory in EA even though discussions about theory within IS are popular. The need for explanation and understanding when using EA was voiced by Radeke [39] without positioning EA as theory. Syynimaa [81] identified the underpinning theory for EA as general systems theory and concluded that future research could investigate the role EA could play to understand organizations from a theoretical perspective, which is a motivation for this study.

In summary, there is a distinct lack of literature that specifically discuss EA and theory. EA is regarded as a mechanism to understand organisations as well as assist with organisational management, organisational change and the alignment of different organisational components. Relevant theory such as organizational theory is often used to assess the impact of EA interventions. However, EA has not been positioned as theory, even though many of the ways in which EA is used support the position that EA is a theory for describing, explaining and understanding organisations.

*2.4. The Zachman Framework for Enterprise Architecture (ZFEA)*

"Most people who come from IT today are thinking of building and running systems and not about engineering and manufacturing enterprises. My argument here is that the end objective is to engineer and manufacture the enterprise, not simply to build and run systems."—John Zachman

The quote is by John Zachman, often hailed as the father of enterprise architecture (EA) [9,81]. Zachman defined EA as a set of descriptive representations relevant to the enterprise, where an enterprise is widely defined as any socio-technical organization [1,2]. As stated, Zachman is particularly known for the Zachman Framework for Enterprise Architecture (ZFEA) that he described as a logical, comprehensive structure or ontology for classifying and organising the descriptive representations of an enterprise in order to engineer (design) the enterprise. Zachman borrowed from an engineering and manufacturing perspective and postulated that any enterprise as a complex socio-technical system needs to be designed and engineered to ensure effectiveness, efficiency and alignment between the different components [10,29,98,99]. Zachman often mentioned that the realisation of the existence of different perspectives relevant to a product applies to an enterprise, and this breakthrough led to the innovative thinking that established the ZFEA [98]. The ZFEA depicts the observation that various engineered objects such as computers, buildings and aeroplanes (designed artefacts) can be

classified according to the fundamental abstractions or interrogatives (ZFEA columns) as well as specific audience perspectives (ZFEA rows). The interrogatives are "What?", "How?", "Where?", "Who?", "When?" and "Why?" (Table 3) and the rows represent business audience perspectives and transformations from a strategic management perspective down to the instantiation perspective of the actual enterprise (Table 4) [10]. The ZFEA (Figure 1) is a 6 × 6 two-dimensional classification schema (enterprise ontology or meta-model) for the descriptive representations of the enterprise without including any process or tooling specifications [100]. Each cell at the intersection of a column and row provides a unique representation or view of the enterprise [10].

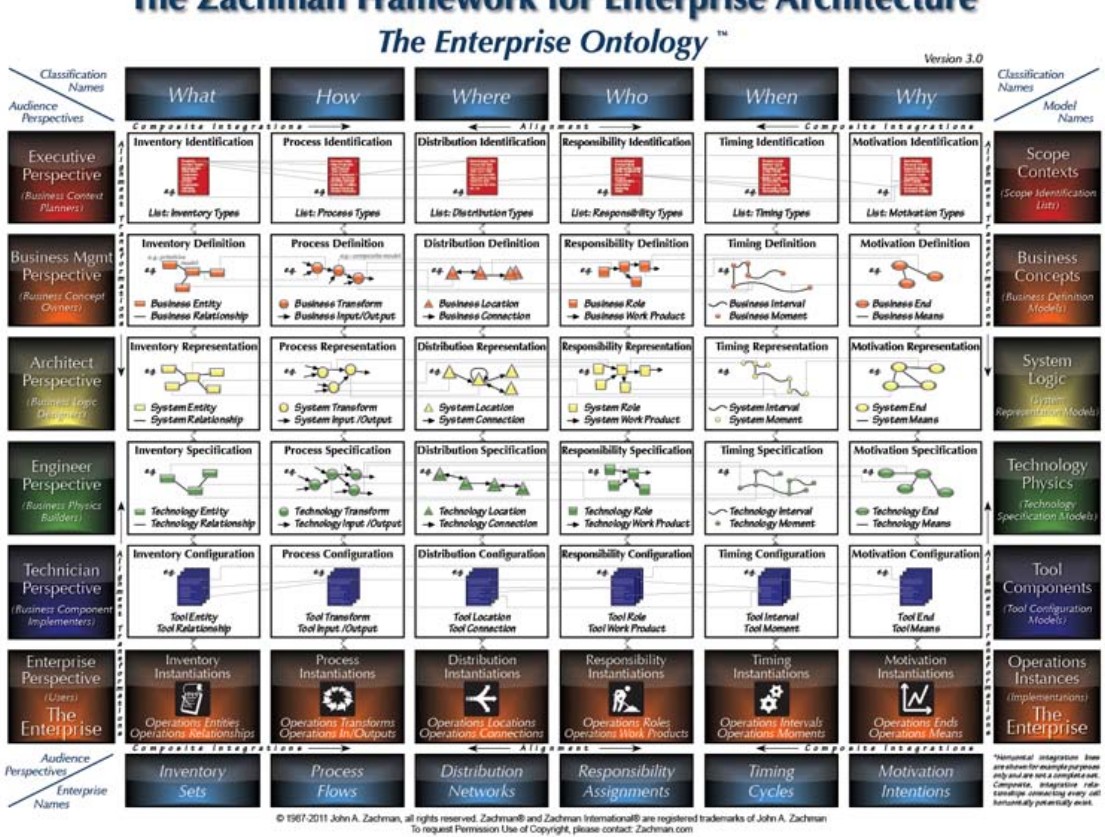

**Figure 1.** The Zachman Framework for Enterprise Architecture (ZFEA) (Reproduced from Reference [10]).

Each ZFEA cell provides the primitive representations that are combined to form the composite models that would represent the designed enterprise given a specific perspective. In the tables below, more information is provided on the rows and columns of the ZFEA.

**Table 3.** Summary of the ZFEA Interrogatives (Abstractions) (Reproduced from Reference [10]).

| Interrogatives (Columns) | Description |
|---|---|
| What? (Data) | What information, business data and objects are involved? |
| How? (Function) | How does it work? (process flows) |
| Where? (Network) | Where are the components located? (network models/distribution networks) |
| Who? (People/Roles) | Who is involved? (workflow models or responsibility assignments) |
| When? (Time) | When do things happen? (timing cycles) |
| Why? (Motivation) | What is the motivation? (business drivers, motivation intensions) |

**Table 4.** Summary of the ZFEA Perspectives (Reproduced from Reference [10]).

| Perspectives (Rows) | Description |
|---|---|
| Executive (Planner) | Contextual View. Defines the limits for all remaining perspectives. |
| Business Manager or CEO (Owner) | Conceptual View. This perspective is concerned with the business itself. |
| Architect (Designer) | Physical View. The architect or person responsible for narrowing the gap between what is required versus what is physically and technically possible. |
| Engineer (Builder) | Represents the perspective of the enterprise engineers interested in "building" or designing the building blocks identified by the architecture. |
| Technician | Represents the perspective of the business technicians such as the database implementers and the workflow system implementers. |
| User (Enterprise) | Represents the perspective of the running/functioning enterprise |

Zachman claimed the ZFEA is "The Enterprise Ontology" or

... a theory of the existence of a structured set of essential components of an object for which explicit expressions are necessary and perhaps even mandatory for creating, operating, and changing the object (the object being an Enterprise, a department, a value chain, a "sliver," a solution, a project, an airplane, a building, a product, a profession or whatever or whatever)—Zachman [10]

The ZFEA does not prescribe any method and is fundamentally only concerned with the meta-model necessary to fully describe any enterprise.

*2.5. Small- and Medium-Sized Enterprises*

The context for answering the second research question of this study is the application of the ZFEA as an explanatory theory in successful SMEs within South Africa. It is anticipated that SMEs, which are widely believed to be important for economic growth and job creation, will create the bulk of new jobs over the next few decades [101–103]. This study adopted the definition for an SME as provided in the National Small Business Act (1996) of South Africa. The turnover and fixed assets depicted in Table 5 have been adapted for inflation to provide a more realistic view of the target population. Thresholds are indicated to be lower than in Europe [104]. As mentioned, this research focuses on understanding and explaining SME growth and transformation. Although SME growth has been studied extensively, research that specifically aims to understand and explain from a theoretical base is lacking [105]. From the literature, it is evident that SME growth studies often lack a suitable holistic view regarding the organisation as a central concept that is being studied, which leads to a lack of insight into the process of growth and an inability to compare organizations during growth [105]. New frameworks or approaches that place the organisation at the centre when studying growth, therefore, have the potential to contribute to understanding and supporting SME growth [105], and, as argued, EA as explanatory theory provides a systems perspective through models and frameworks, often as visual representations of an enterprise, its components and the interaction between its underlying elements [106].

**Table 5.** Schedule 1 from the National Small Business Act of South Africa that Defines SMEs within the South African Context. The table is reproduced from the act.

| Enterprise Size | Number of Total Full-Time Employees | Annual Turnover (in Million Rand (ZAR M)) | Gross Assets, Excluding Fixed Property < 1 Million South African Rand (ZAR M) |
|---|---|---|---|
| Micro | <5 | <0.4 | <0.26 |
| Very small | 10-20 | 0.4 to 2 | <1 |
| Small | 20–50 | 2 to 64 | <12 |
| Medium | 50–200 | 64 to 128 | <46 |

*2.6. The Zachman Framework for Enterprise Architecture as an Explanatory IS Theory*

In this section, we present previous work where we mapped the ZFEA to Gregor's theory taxonomy as well as the structural components of the theory to determine if the framework can serve as an IS theory [107]. We first explain the motivation for using the ZFEA in the next section followed by the section where we apply Gregor's method to classify the ZFEA.

2.6.1. Motivation for the Adoption of the Zachman Framework for Enterprise Architecture

EA originally developed because of the need to integrate computer systems or information technology (IT) into organisations in such a way that strategy and business processes are supported effectively and efficiently [22]. The fundamental systems perspective that underpins EA was adopted by various disciplines, but all had similar objectives that include the ability to understand and manage complexity and change, as well as to ensure organisational alignment between different components [29,56]—a concept often termed "business–IT alignment" within EA literature. The definitions of exactly what EA entails evolved with the discipline, but nowadays most of the accepted definitions include notions such as that EA is the continuous practice of describing the essential elements of a socio-technical organisation and their relationships to each other and to the environment, in order to understand complexity and manage change [22,23].

As mentioned before, Zachman was one of the founders of EA and his departing perspective was that any enterprise is a system that needs to be designed and engineered for alignment and effectiveness, and in order do that, EA is required. Zachman defined EA as a set of descriptive representations relevant to the enterprise that ensures through design that the business with its technologies and resources, as well as the current and future purposes of the organisation, are aligned [108]. According to Zachman, the description of the enterprise or the architecture built using the ZFEA schema would necessarily constitute the total set of descriptive representations that are relevant for describing the enterprise [10,108]. Since its inception, several different EA definitions and frameworks developed from different perspectives and domains, each with unique applications and practices, but the fundamental departure point as originally developed by Zachman remains distinguishable.

EA frameworks, as representations of the holistic enterprise from which to address the problem of managing change, complexity and alignment between enterprise elements, are central to the EA discipline [109,110]. In a complex environment of constant change, EA frameworks assist practitioners to make sense of complexity and drive sustainability [56,109]. EA frameworks define the rules, principles and techniques used in EA [111]. Frameworks may have different viewpoints and may be used in different contexts. Some frameworks focus on architecture descriptions and others on the EA method; however, all frameworks need to describe and represent an enterprise (the *architecture* in enterprise architecture) and all frameworks, therefore, have descriptive representations as defined by Zachman.

For a study that argues that EA may serve as an explanatory theory, an EA framework that fundamentally represents the total set of enterprise elements is required to enable comparison. Even though certain elements may be less pronounced in certain contexts, it does not mean they do not exist. The same way that all elements on the periodic table exists in nature as a total system,

the enterprise as a total system has a definitive number of elements, whether explicitly represented or not [10]. If we want to explain aspects of businesses, such as SME growth, in more than one context or for more than one business, we need the ability to compare structural components. It is this ability of an appropriate EA framework that makes it so useful as an explanatory IS theory and management tool.

Zachman himself placed a lot of emphasis on the fact that EA needs to represent all aspects of an enterprise and the ZFEA was specifically developed to fulfil this purpose. Zachman coined what he calls "The Enterprise Laws of Physics" that include "The First Law of Enterprise Ontological Holism", namely that "Every Cell of the Enterprise Ontology exists. Any Cell or portion of Cell that is not made explicit is implicit which means that you are allowing anyone and everyone to make whatever assumptions they want to make about the contents and structure of that Cell" [22,112]. The ZFEA is therefore explicitly described as an ontology or theory of a structured set of elements of an enterprise [10]. These fundamental departure points, as well as the thorough focus on the description of the enterprise, serves as the motivation for using the ZFEA in this study. The ZFEA is not the only framework that can be used but was chosen for its originality and to provide a basis from which to test the hypothesis that EA could be a theory. Should the hypothesis prove true, it should be possible to build a new framework or use another framework in the same context as long as all the enterprise elements and their interrelationships are represented in the framework.

### 2.6.2. Classifying the ZFEA using Gregor's Theory Taxonomy and Structural Components

In the first step, positioning the ZFEA as IS theory, we classified the ZFEA as one of Gregor's theory types by considering the primary goals of the theory [19]. As stated for this section of the study, we adopted the method proposed by Gregor herself to demonstrate the use of the theory taxonomy [19]. Using the taxonomy of theory types, the following analyses represent the classification of the ZFEA:

- Type I: The ZFEA has as a primary function "analysis and description" or providing a complete representation of the enterprise through framework elements (Type I Theory).
- Type II: The ZFEA does extend beyond "analysis and description" by providing explanations (i.e., "say what is, how, why, when and where" (as specified by Gregor [19])). This is done through the perspectives and rows as well as relevant interdependencies. The ZFEA does not aim to predict with precision, which is typical of a Type II Theory that explains but does not predict with precision.
- Type III: The ZFEA does not say "what is and what will be" and nor does the ZFEA have testable propositions (characteristics of a Type III Theory).
- Type IV: The ZFEA aims to support causal explanations but does not pose direct quantitative causal explanations (the nature of Type IV Theories).
- Type V: The ZFEA specifies no process or tooling specifications and is, therefore, not a Type V Theory (design and action).

Since the ZFEA is a descriptive structural representation of an organisation and all its elements with their relationships directed at providing understanding and insight, the ZFEA can be regarded as a Type II or explanatory theory that "says what is, how, why, when and where". A Type II Theory furthermore "provides explanation but does not aim to predict with any precision. There are no testable propositions" [19].

As a second step to position the ZFEA as explanatory IS theory, we used the second part of the method proposed by Gregor [19] and analysed whether it is possible to identify the necessary theory components of an IS theory. This analysis is presented in Table 6.

**Table 6.** ZFEA and Theory Components of Gregor [19].

**Theory Overview:** The ZFEA is an ontology, a $6 \times 6$ two-dimensional schema and a structure that is descriptive in nature. The architecture of a specific enterprise that was developed using the ZFEA schema or ontology would necessarily constitute the total set of descriptive representations that are relevant for describing the enterprise.

| Theory Component | Instantiation: ZFEA |
|---|---|
| Means of representation | Conforms: Words, tables, diagrams, the ZFEA is a $6 \times 6$ matrix consisting of a diagram and tables with accompanying descriptions. |
| Primary constructs | Conforms: The complex object is the enterprise with its strategy, technology, processes, people, roles, etc. A holistic view is displayed. Objects are viewed from different perspectives and interrogative abstractions. |
| Statements of relationship | Conforms: Relationships between the audience perspectives and interrogative abstractions are specified as transformations, and, within each cell, primitives have predefined relationships. Relationships in the ZFEA (and EA in general) are very comprehensive, i.e., dependent, associated, linked, bi-directional or multi-directional, etc. |
| Scope | Conforms: The scope is specified by the degree of generality of the statements of relationships. The ZFEA is a general schema that aims to provide a holistic view of any enterprise or engineered (complex) object and a very high level of generality is proposed. Generalisation was part of the ZFEA development as the schema is derived from observing many different objects and industries. |
| Causal explanations | Conforms: The ZFEA attempts to give statements of relationships among phenomena (represented by the rows and columns in the matrix). The ZFEA aims to support causal explanations. |
| Testable propositions (hypotheses) | Does not conform: An explanatory theory typically does not conform to this component. Statements of relationships between constructs that are stated in such a form that they can be tested empirically are not present. Zachman states that the model should not be applied deterministically but that it is an ontology that is repeatable and testable (such as the periodic table), however, there is not yet evidence of the ZFEA being implemented in this manner. |
| Prescriptive statements | Does not conform: An explanatory theory typically does not conform to this component. Statements in the theory specify how people can accomplish something in practice (e.g., construct an artefact or develop a strategy). This is somewhat supported by the ZFEA as the purpose of the ZFEA is to model an enterprise by using the interrogatives and perspectives, however, a detailed process or method is not supported. |

From the analyses using the method of Gregor, it is possible to argue for the ZFEA as an explanatory theory that aims to "provide an understanding on how, when and why an occurrence took place based on causality and argumentation" [19]. Zachman also describes the ZFEA as an ontology and a structural schema that aims to be a repeatable and testable description of an enterprise. If we consider the requirements of an explanatory theory, we could argue that the ZFEA aligns as it intends to provide insights into the how, when and why of an enterprise. The ZFEA is a framework and ontology that provides a holistic view of an enterprise when all the elements are populated, and the ZFEA also provides insights given the specific perspectives (or rows). If each row, column and cell in the architecture is considered for a specific enterprise, the impact by another row, cell or column is distinguishable, e.g., the technology supports the business processes, the business processes the applications and the applications, the strategy. Changes in any model will, therefore, affect the other models. Changes in strategy (cause), for example, will have an impact (effect) on the rest of the enterprise. All components form part of the whole to provide context. The scope is defined by the subset or component being designed. It can be argued that the main goals of the ZFEA are aligned

to the goals of an explanatory theory since both aim to provide insight, understanding and causal explanations, as well as indicate relationships among components.

After we theoretically positioned the ZFEA as explanatory theory, we have answered the first research question. To answer the second question, we use the ZFEA as a theoretical basis to analyse successful SMEs towards understanding and explaining successful growth and transformation through identified focus patterns. The next section presents the research method and data collection.

## 3. Research Method and Data Collection

### 3.1. Case Studies

The purpose of this part of the study is to investigate whether the ZFEA effectively serves as an explanatory theory that would assist to *understand* and *explain* SME growth and transformation. Towards this purpose, we adopted a carefully designed multiple case study method. Case studies are especially applicable in instances where the research aims to understand "how" questions and when the phenomenon being studied is rooted in a real-life context where the object of study and the phenomenon being studied are not separable [113–115]. Case studies are further applicable when the researcher has no control over the events being studied and this research can be used for theory building and testing [113–115]. Multiple case studies normally provide a stronger base for theory building and enable a broader exploration of research questions [114]. In this study, the ZFEA was used as a basis to analyse the management focus in the case study SMEs. Within the case studies, the person directing the organisation and the organisational context within which this management takes place are bound together and cannot be separated, nor did the researchers have any influence over the thinking and actions that took place [113,114]. With regard to the strengths of case studies, interpretive case studies are applicable for theory building and testing and also assist in providing a deeper and better-contextualised understanding of the phenomenon being studied, which is particularly relevant for this study [113]. With regard to the major known disadvantages of case study research—namely, that it is difficult to generalise to a broader context using qualitative case studies, controls that may often be lacking and subjectivity [113,116]—particular care was taken so that these disadvantages were addressed during the execution of the actual research. Case study research provides a rich empirical description of a particular instance of a phenomenon [114] and single as well as multiple case study approaches are widely used research approaches for studying IS and EA in organisations [117,118]. Several data collection methods apply to case studies including interviews, documents, questionnaires and observations [113,119]. Interviews are one of the most used methods for collection of data in case study research [116,120]. Interviews are an efficient manner to collect rich empirical data, especially when the phenomenon being studied is infrequent and episodic [114], with semi-structured interviews well-suited for gathering data in a specific context about someone's experience [120].

Case selection is an important step in case study research [119]. When building a theory from case study research, the aim of the study needs to be established beforehand in order to assist in identifying the type of organization to be approached [115,119]. It is a faulty assumption that case studies should be representative of some population, especially in instances where a theory is being developed and not tested, and in such instances theoretical sampling can be proposed as an alternative [114]. For this study, the target population for cases consisted of several SMEs within the South African context and the main data collection methods adopted were interviews and observations as discussed in the next sections. Using collaboration networks, we adopted purposeful sampling and identified seven different successful SMEs. What is deemed successful is detailed in the following sections and criteria included the number of employees and/or turnover, as well as the number of years operating. We identified 11 participants from the 7 SMEs that were interviewed, all founders and executives currently still involved in the SMEs.

### 3.2. Data Collection and Analysis

The SMEs used for data collection in this study are country-specific, namely, South African. Business circumstances within South Africa are challenging due to a shrinking economic base, junk status, crime and ineffective structural support [121,122]. SMEs within South Africa are generally defined as enterprises with fewer than 200 employees and an annual turnover of less than R30M (<\$10M) (refer to Table 5). The most common way to study organisational growth or size is through turnover or number of employees [105,123]. In this study, the number of employees and/or turnover were used as a measurement of size; thus, inclusion into the study—and a successful SME in South Africa—is defined as an organisation that has been in operation for more than three years.

The data were collected from 11 participants in 7 different cases; all are successful SMEs operating primarily within South Africa. The interview participants were all owners of the case SMEs and drivers for business success. The whole point to enable appropriate business focus and comparisons is to speak to people who were responsible for the growth. The basis for selecting the participants is also because we wanted to interview people directly involved and responsible for the business start and growth, an aspect which supports the uniqueness of the study. All participants were involved from the beginning and focused in a certain pattern on organisational elements. The SMEs selected for the multiple case studies offered a combination of products and services. Services companies are more income state than balance sheet focused, making gross assets less of an indicator for size. The SMEs furthermore all had turnover and/or employment in line with Table 5.

In this study, the knowledge and experience of SME owners/managers during start-up and growth were pertinent, and interviews were, therefore, used as the main data collection method and 11 semi-structured interviews were conducted mainly with founders and executives currently still involved in the SMEs. These individuals were selected because they were and still are in most cases responsible for directing and enacting the growth of their businesses.

The interviews were augmented with observations; own notes from, for instance, site visits and, where applicable, written documents [119]. The participants, as well as additional data sources that were considered, are included in Appendix B. Given the central setting of interviews in the data collection strategy, the design and execution of the interviews are important aspects of the study and it is recommended to interview multiple participants, as it is unlikely that multiple participants will engage in impression management or retrospective sense-making [114]. These participants could be from various hierarchical levels within organisations, or even role players from different organisations or geographical areas [114]. The premise of the ZFEA as an explanatory theory was used to develop questions and Appendix A provides the open-ended interview structure, with the questions and rationale for the questions [107,115,119].

Data collected were non-numeric qualitative data, for which there are some guidelines but no firm set of rules in terms of analysis [124,125]. Rigour in interpretive studies requires the researcher to clearly show how interpretations of the data have been achieved [125]. Thematic analysis is a helpful method to create a link between data from interviews and the interpretation of the data [120,125]; conversational analysis is a suitable data analysis approach for interviews. Both methods were adopted for this study [116].

All conducted interviews were recorded and transcribed in order for the text to be analysed. Guidelines for transcribing interviews so as not to lose the "richness" of the interview [124] as well as practical rules regarding the transcripts are recommended practices for interpretive case study data analysis [126]. After transcription, all the data were encoded using thematic analysis based in EA and the ZFEA. The coding was subsequently verified individually and during a discussion session by researchers with an in-depth knowledge of the ZFEA. Atlas.TI was coded with ZFEA elements as a base reference as well as relevant themes that emerged from the interviews. Words were set as the smallest and paragraphs/concepts as larger units of analysis. Subsequently, the response of each person per case was mapped to the different elements of the ZFEA and counted. To corroborate the mapping, a second mapping was done in a table format, with the mapping done in excruciating detail, including

the original language and translations, to corroborate the mapping done in Atlas.TI. This step was executed so that the logic of the mapping can be shown and compared for future studies in different contexts. Based on the mapping per case, in sequence and the total mapping, clear patterns could be identified.

For this study, the following transcription and analysis principles were applied:

- The interviews, including the mannerisms of the interviewer and participant, were transcribed. This enabled a richer analysis of the text and allowed for a better understanding and negating of interviewer influence.
- Replication logic is an important aspect in multiple case studies, and each case should be seen as a standalone experiment and analytical unit [119]. Each case was analysed separately and afterwards, all the concluded analyses were combined to form an overarching view across cases.
- Following the transcription, the text was imported into an analysis tool and analysed per case based on the topic under discussion, with special emphasis on the questions and explanations of questions. The unit of analysis was the person being interviewed as representative of the organisation being studied. The unit for analysis in the transcribed text ranged from a single word as the smallest unit to a paragraph or concept as the biggest unit.
- The sequence of analysis followed a chronological process based on the order in which the interviews were completed and transcribed, starting with the first and ending with the last.
- The structured representation of the organisation provided by the ZFEA served as the basis to identify possible patterns of focus. The cases were analysed on their combined responses to identify how EA knowledge or knowledge about the holistic organisation and its underlying parts relevant for growth. During the analysis, the context for each case was used to determine the meaning of words, especially where different words were used to describe the same concept.

Figure 2 depicts a sample of the coding used using Atlas.TI as well as colour-coded transcriptions to detect the focus patterns.

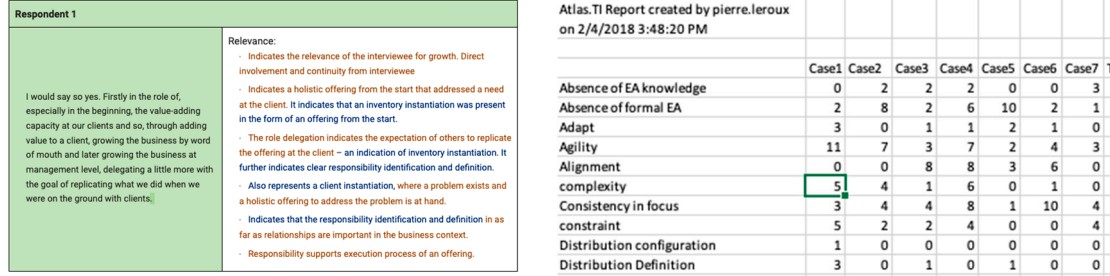

**Figure 2.** Samples of thematic coding of case study data.

## 4. Results and Findings of Case Studies

In this section, the results and findings from the multiple SME case studies are presented. The results are presented according to the interview questions, grouped into three logical sections, to allow for analysis across cases. Section 4.1 presents a demographic overview of the participating cases, the eligibility parameters, and the participants' responsibility for growth in their organizations. Section 4.2 focuses on practical aspects related to EA in the participating cases: knowledge about EA, business transformation and EA frameworks in use. Section 4.3 presents the main results with regard to the ZFEA aspects or elements of the SMEs during start-up, growth and at present.

### 4.1. Demographic Overview

Regarding age, the case study SMEs started business from 1997 through to 2013. The youngest organization was four years old and the oldest 20 years old. With an average existence of 11.14 years,

there was evidence that the cases were well-established as going concerns. All organizations adhered to the "time in existence" definition for a successful SME for this study.

The industries differed across all seven cases. The industries in which the cases operated were IT, business consultation, telecommunications, auditing and accounting, rigging and lifting, software development, financial and legal services, and construction consulting. The offerings of the cases were a mix of products and services, with three cases providing products and four providing services. The products and services were diverse in nature, making for an interesting comparison across focus areas.

Employment numbers ranged from nine to 120 employees. There were no indications of a correlation between the age of the organization and the number of employees. The oldest organization had 20 to 30 employees at any given time and the youngest had 20. The highest employee numbers of 110 and 120 were nine and 10 years old, respectively. All SMEs but one fell within the stated employment parameters of between 20 to 200 employees for SMEs. The case with nine employees, however, actively drove low employment and qualified under the turnover parameter; thus, it was not excluded from the study.

All cases complied with the parameters of an annual turnover of between R2 million and R128 million. Due to the sensitivity of this information, the exact turnover was not enquired and is not relevant for this study.

All the participants that were interviewed from the seven cases either started the SMEs or were involved from the beginning. Due to their involvement, the participants as representatives of the participating cases provided a suitable basis for analysis from which to address the aim of the study.

*4.2. Enterprise Architecture Knowledge and Exposure*

There was a mix of participants knowledgeable about EA. Three cases displayed a solid understanding of EA, while three did not have any understanding of or previous exposure to EA. In one case, the one participant indicated having a good understanding regarding EA but the second did not. The split between cases knowledgeable and not knowledgeable about EA was 50 per cent. In the instances where knowledge about EA was evident, they could provide concise definitions of EA, however, they defined EA in the domain of organization or enterprise as a holistic system with an understanding of its underlying elements. They, therefore, approached EA from the classical systems perspective and not from a framework or methodological perspective.

The participants defined EA by responding that it is necessary to adapt their organizations quickly in order to stay relevant. This question was asked to determine how participants see the process, focus and management of organisational change and transformation. It helped to identify to what extent a structured manner for managing change applied, and to determine the impact level on certain enterprise elements during change. The need to transform the enterprise in an intelligent manner was evident in all cases, with business agility acknowledged as necessary. Phrases like "definitely", "you have to be agile", "planning the softer side of change" and "being forced to adapt" all indicated the relevance of change.

Across cases, there were indications of an understanding regarding the interaction of enterprise elements, especially in the cases familiar with EA. Different wording and nuances described the interaction between the enterprise elements and the angle of interaction differed, but interaction between elements was indicated to be relevant for success. Phrases like "interaction of building blocks"; "driving building blocks back and forth"; "how you structure the business in the context of the environment"; and "the alignment between what a business wants to do, what resources are available to do it with and what process it follows to achieve these goals" all indicated an understanding of the interaction of different aspects of the enterprise to form a whole.

There was a lack of formally documented EA in all cases. Even though all participants had an idea and mental roadmap for transformation, not one case formally documented EA. Half of the participants knew of EA and, as a result, could explain their thinking by referring to an EA framework. However,

the participants not familiar with EA also took cognizance of the impact of enterprise elements on one another during change, indicating that some form of systems thinking was in place in all cases. Phrases like "an understanding", "study up" and "establish the processes to improve the end product" all indicated the relevance of change and understanding the impact of change. There was also an indication that the process of enabling and executing change was relevant, with phrases like "planning for change, adapting to better the product, and adapting the process" indicating this.

There was a lack of adoption of formal EA frameworks during enterprise transformation across all cases. There were indications that all cases considered the impact of enterprise elements on one another during change, but this was a cognitive consideration for all cases and there was no evidence of formal models or frameworks being applied. Models provided by case 1 were more a methodology and process than an EA framework. There was evidence that, in the cases familiar with EA, thinking was guided by defined principles. Words and phrases like "framework", "checklist", "analyze where you want to go" and "an understanding of the organization" all indicated a good grasp of the organisation, its contents and surroundings. One participant argued for the use of a checklist when changing the organization and the assistance this would provide to owners of new organizations. Even though some cases acknowledged a checklist, none of the cases actively used a framework during change, which indicated that the complexity was low enough to manage change without formal documentation.

The environment was identified to play a big role in the enterprises, and there was evidence across cases regarding the impact of the environment on the business and business change. Discussions about "your own and your client's environment", "existence in context of other stakeholder environments", the "influence of external factors" and "impact of legislation" are examples of the environment's influence on the enterprises. The literature supports the relevance of change and the impact of the environment on the organisation, and the evidence that it was the same for the participating cases is not surprising. During enterprise transformation, there was evidence across cases that an understanding regarding the impact of the environment on the enterprise existed, and that the impact was considered during planning. Even though not documented, stakeholder identification and participation were evident across cases, with the impact on stakeholders during change well understood and managed. Over time, the participants seemed to understand the process of change better, with the identification of internal impacts and controls evident.

Across all cases, growth was seen to be organic and no specific stages or common inflexion points were evident. During enterprise growth, there was acknowledgement of increased complexity. During periods of growth, there was evidence that the focus moved away from the enterprise, executive and management perspectives, more towards the architect, engineer and technician perspectives of the ZFEA. The discussions indicated a focus on enterprise design during transformation. Despite the focus on the architect, engineer and technician perspectives, there was a measure of consistency in the focus on the enterprise elements that were relevant from the start as well. In line with the third and fourth laws of enterprise ontological holism [107], the overseers of growth were indicated to have a good understanding of rows 1 and 2 to maintain business system stability during growth.

*4.3. Focus Patterns*

The third section of the interviews focused on the detection of focus patterns based on the ZFEA during the start-up and growth of the enterprises.

A certain level of knowledge regarding the industry was evident across cases, with phrases such as "actually contracted" and "gap in the market" indicating this knowledge. Different company structures and roles, representing the "who" interrogative, different processes, inventory items localities, cycles, motivations and so on, were discussed. These, with the help of an underlying classification mechanism like the ZFEA, could be analysed for trends.

During start-up, there were clear distinctions and allocation of responsibility between roles. Even though founding members often played different roles, the role definitions were clear. There was evidence that identification, definition and delegation of roles and responsibilities took place early

in the life of the enterprise. Phrases like "delegating with the goal of replicating what we did on the ground" and "acted as", together with the discussion of responsibilities in the context of different roles, indicated a clear distinction of roles and responsibilities.

### 4.3.1. Start-up Focus Patterns

- Contrary to what was expected, the focus during start-up was not on the strategic perspective (top row) of the ZFEA but in the instantiation row or the enterprise perspective. There was evidence of the strategic perspective as well and transformation happens upwards through alignment between the enterprise, executive and management perspectives of the framework.
- All cases indicated a pragmatic and intuitive approach to enterprise start-up. The terms used by the participants included "service", "people" and "offering", or a description of a specific process, all of which indicated instantiation, but the instantiated processes, offerings or roles could be explained alluding to the executive perspective. Across cases, the value and role of people were evident. From a ZFEA point of view, the perspectives on people differed depending on the stage of growth, but people, as a general concept, are very relevant for enterprises in the South African context.

### 4.3.2. Enterprise Growth Focus Patterns

- There was an indication that all cases had an intention to grow and growth occurred organically in all cases, with no formal identifiable trigger points. However, growth drove complexity across all cases, which increased the need for agility and the ability to adapt. EA thinking and an understanding of the impact of enterprise elements on one another were indicated to be relevant for growth, with participants appearing to actively try to understand the impact of growth on the enterprise. Design aspects of the ZFEA were showed to be more relevant during change.
- From a growth perspective, there was evidence of participants having oversight over the complete enterprises from a clearly defined point of responsibility. There was also evidence of direct involvement from the participants in the physical execution (instantiation) of their business offering in the beginning. In many instances, the role the participants fulfilled was to execute the offering at an enterprise or instantiation level, with clear alignment between the executive and instantiated responsibility. The concept of instantiated relationships as responsibility type was evident across cases.
- Across cases, there was evidence of a clear differentiation between responsibilities viewed from the ZFEA enterprise, executive and management perspectives. There were indications that one person typically fulfilled more than one role and acted from more than one perspective to ensure growth. There was evidence that the participants understood the difference between roles and alternated between roles as required. The environment created the context within which the organisations provided clearly defined offerings, with direct involvement by the participants in the identification, setting up and execution of key processes during their role in growth.
- The focus patterns that had been identified as relevant during enterprise start-up were still relevant, but additional elements of focus became visible as enterprise complexity increased. During growth, a lower granularity of focus was evident, with the architect, engineer and technician perspectives evident across interrogatives. References to, for example, "systems", "understanding the impact", "skill sets" and "process set-up" indicated this focus.

### 4.3.3. Normal Operations Focus Patterns

- During normal operations, the analyses of focus points provided valuable insight across cases. The patterns that emerged during the previous two angles of analysis stayed relevant, and the focus during the "current" operations served to bring the viewpoints together. During the analysis, a lack of formal EA practices was confirmed for all cases, but there was evidence that all cases

had insight into the impact of enterprise elements on one another, which confirmed the earlier findings in the analysis. Following growth, the focus of cases was indicated to include the complete enterprise, with a focus on the initial elements (inventory, process, responsibility) along with the added realisation that more detailed and different viewpoints were relevant to maintain the enterprise. EA thinking, understanding the impact of growth and the need for planning were, thus, confirmed.

- All cases (with and without knowledge about EA) indicated that they believed understanding the interaction between an organisation's elements and the impact of elements on one another contributed to their organisations' success.

- As stated, start-up organisations focused on the instantiation/enterprise row (Row 6) and as the business grew, transformation upwards was identified in all cases. There was evidence that the relationship and alignment between the enterprise, executive and management perspectives provided the platform for a stable business system from the start. All cases had a correlation and alignment between row 6 (instantiation) and rows 1 and 2 (executive and management). Transformation from row 6 upwards could assist in alignment, as the instantiated interrogative is often practically described and makes for easier intuitive alignment.

*4.4. Consolidation: ZFEA-Based Focus Patterns of Successful SMEs during Start-up, Growth and Transformation*

All cases in the study adhered to the definition of qualifying SMEs in terms of turnover and/or employee numbers, as well as age. The cases operated across different industries, and all the participants were directly responsible for business growth or success. Multiple industries and offerings (products and/or services) provided a diversified base from which to draw comparisons and conclusions. All cases studied were indicated to be good examples of job creation, business growth and successful SMEs in the South African context.

All the participants were directly responsible for and involved in enterprise growth and success, as well as identifying, setting up and execution of key processes during start-up and growth. All participants had been involved with the enterprises from the start or at a very early stage, making the comparison between cases relating to focal points during start-up possible. All participants were still part of the daily management of the enterprises, making the comparison between cases regarding the focal points during growth and at present possible. There was an indication that all participants had a certain level of knowledge about the industry or the problem/opportunity that existed at the start of the enterprise.

There was evidence of environmental impact on the enterprises during start-up, firstly as the context within which to deliver an offering and secondly as an influencing factor on the enterprise. Across cases, the environment was the basis for creating context regarding the business offering, with either a problem or opportunity that existed in the environment serving as the foundation for context.

All cases had a complete absence of formal EA frameworks, practices and documentation. This indicated a correlation with the thinking in reference [106] that EA modelling becomes more relevant as the size of the organisation drives complexity to a level where it cannot be managed without documentation.

Cases with a background in EA were able to use EA to explain the concepts of focus using their own frame of reference. There was evidence that all cases had substantial knowledge of the holistic enterprise, the underlying elements, and the interaction between the elements.

There was an absence of definable growth stages or measurable points of inflexion for growth across all cases. Across cases, growth seemed to be organic, with different tempos of growth. There was an indication that the environment often played an influencing role in growth.

There was evidence that all cases could prioritise functions at a given time, with the configuration and formality of processes growing over time and with the business. Although process execution occurred at an operational level, a total view of the business existed from the beginning. There was direct involvement by the participants in the identification, setting up and execution of key processes during start-up and growth. The current focus for every case indicated a correlation with the focus earlier in the organisation's life and there was a consistency in focus.

All cases seemed to have the holistic organisation and key elements in place from the start. Adaptions were made where needed, but the key focus of the organisations remained constant.

The concept of people understanding the relevant focus of an organisation supported the notion that a suitable conceptual support framework may assist to guide future organisations. Across cases, patterns of focus emerged during start-up, enterprise growth and transformation, as well as maintenance.

### 4.4.1. ZFEA Perspectives or Rows

Start-up: During start-up, there was a pragmatic focus on the bottom instantiation row or enterprise perspective. Contrary to expectations, the focus on enterprise elements did not originate in the strategic "why" interrogative, nor from the executive perspective, but was rather an upwards transformation from an instantiation in row 6 of the ZFEA.

Growth and Transformation: During growth and transformation, all participants were aware of increased complexity of the enterprise as well as the impact of a challenging operating environment. All participants indicated an awareness of enterprise elements influencing each other. Additional focal elements emerged, and a lower granularity of focus was evident, with the architect, engineer and technician perspectives evident across interrogatives with transformation migrating upwards towards the strategy level.

Normal Operations: The focus of all cases included the complete enterprise, with a focus on the initial elements (inventory, process, responsibility) along with the added realisation that more detailed and different viewpoints were relevant to maintain the enterprise.

### 4.4.2. ZFEA Columns or Interrogatives

All cases indicated the entire business system to be operational from the beginning, although with a focus on some key elements. Three elements often recurred and were more explicit than others, namely, the inventory (what), process (how) and responsibility (who) instantiation that existed from the start, but in the context of a need or problem in the environment. There seemed to be a common theme amongst all cases in that the offering (inventory) in the context of a problem, with the key processes and responsibilities needed to execute on the offering, was in place from the start.

The environment often affected the inventory, delivery process or responsibilities of the organisation, which needed to be adapted to stay relevant.

There was an indication that the speed of growth was not constant and that the environment and internal alignment played a role regarding the speed of growth. Internal alignment of the organisation in the context of addressing a specific need in the environment seemed to enable growth.

All cases had a complete offering in the context of a problem or opportunity identified in the enterprise environment. All cases had the key processes in place through which to deliver the offerings. The key processes differed from case to case depending on the offering, enterprise environment and enterprise need. Interestingly, the process primitives seemed to have some overlap but also had significant differences. Some supporting processes were similar. Examples of key processes ranged from personnel appointments, quality assurance, marketing, training and customer service, to project execution. The correlation resided in the roles these key processes played in the execution of the offering/inventory for the case in question.

All cases had direct control over the needed responsibilities to execute the processes of delivering the offerings. The lists of responsibilities and their interactions with the process had nuances based on the need of the individual enterprise, but the underlying pattern was similar. There was a recognition of different roles although the founding members often fulfilled more than one role.

### 4.4.3. ZFEA Transformations

For all cases, there was evidence of consistency in focus from start-up to the current management and running of the enterprise. Slight adaptions were evident where relevant to adapt the enterprise, but the underlying pattern of focus across inventory, process and responsibility seemed to be constant.

During start-up, the focus on enterprise elements did not originate in the strategic "why" interrogative, nor from the executive perspective, but from an instantiation enterprise perspective in row 6 of the ZFEA. An upwards transformation from the enterprise perspective—with a specific focus on inventory, process and responsibility instantiation—was evident. Motivation seemed to be less of a factor and influence than would have been expected. During start-up, there was limited focus on elements from the architect, engineer and technician perspectives.

All cases displayed a certain level of insight regarding the impact of the environment on the enterprise. There was evidence of an appreciation that the environment of the enterprise had an impact on the enterprise and that the enterprise needed to adapt. In all cases, the complexity and automation of the offering increased over time and with growth. However, the minimum viable inventory instantiation to service the need was in place from the start.

During growth, the enterprise adapted as needed to keep the alignment between enterprise elements and between the enterprise and the environment. During enterprise growth, the concept of instantiation as a starting point for adapting the enterprise, with upwards transformation into other perspectives, seemed relevant. There seemed to be an understanding regarding the impact of enterprise elements on one another during growth, even though there was a lack of formal EA modelling in all cases.

There were indications that the interaction between enterprise elements during growth assisted with maintaining the structural integrity of the enterprise by keeping all the elements aligned. During growth, the complexity of the enterprise increased substantially. There seemed to be an increase in focus on the engineer, architect and technician perspectives of the ZFEA during growth. Process and responsibility seemed to be better defined and more pronounced during growth.

Figure 3 indicates the focus areas of the combined cases during start-up, growth and transformation. When combined, representation of the complete enterprise perspective was evident, but timing, motivation and distribution instantiation were not equally relevant for all cases. As specified by the ZFEA, every cell in the enterprise ontology exists, whether explicitly or not, but during start-up and growth, different focus points emerge.

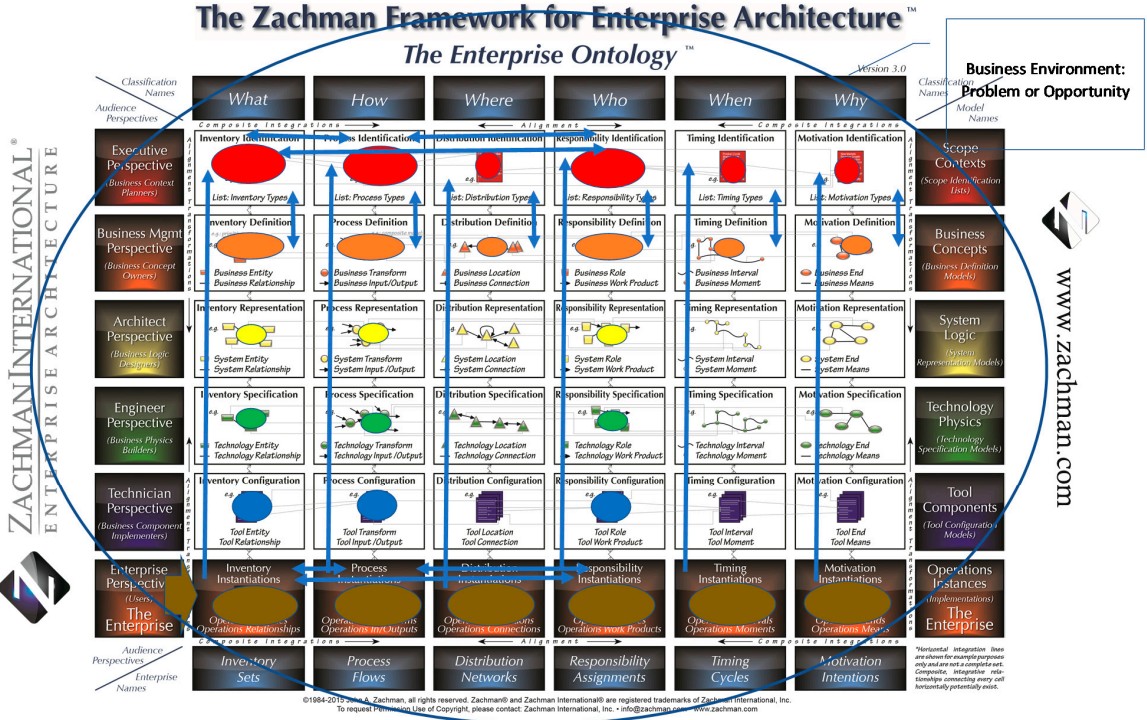

**Figure 3.** Enterprise Elements Focused on during Start-up and Growth for Combined Cases Depicted on the ZFEA.

## 5. Discussion

This paper reports on a study that was concerned with understanding SME growth using the ZFEA as a theory. To execute the study, we first had to consider IS theory and determine whether the ZFEA should be considered an explanatory IS theory. Secondly, we used the ZFEA as the basis to analyse and understand SME growth through the detection of focus patterns based on the ZFEA.

### 5.1. Theory for Enterprise Architecture

One of the main contributions of the study is positioning the ZFEA as an explanatory IS theory. This position is unique as the descriptive representation or the architecture of EA itself as represented by the ZFEA has not been considered as a theory in literature before to our knowledge.

Literature about theory within and for EA is limited, and discussions primarily focus on analysing EA's impact and implementation from organisational theory perspectives [30,65,76,77,83,93,94]. These perspectives adopt the viewpoint that EA is an organisational intervention or practice, which is part of the practical and procedural aspects of especially EA frameworks such as TOGAF [97]. Some discussions about EA refer to the fundamental departure points and argue that EA adopts a systems theory paradigm [80,81]. Zachman originally adopted a systems engineering perspective, and this perspective is supported by the results of this paper as systems theory also supports understanding in general. However, since EA's inception, there was a shift in that EA scholars and practitioners mainly focused on the practical and pragmatic aspects of EA execution. More recent research again refers to the original foundations and the need for understanding the role EA could play as theory. Radeke [39] identified the need for explanation within EAM and Syynimaa [81] identified the underpinning theory for EA as general systems theory and concluded by stating that this may allow for EA to be an explanatory theory, which supports our work that uses Gregor's taxonomy and method to classify the ZFEA as an explanatory theory [107]. According to Gregor, an explanatory theory is used to say " ... what is, how, why, when and where: The main aim is one of explanation and to provide understanding. The theory provides explanations but does not aim to predict with any precision". Gregor furthermore asserts that explanatory theory is a building block of the more advanced theories, namely, theories of prediction, explanation and prediction, and design and action because it is necessary to understand and explain before one can predict.

### 5.2. The Application of the ZFEA as Theory to Understand SME Growth

The second part of the paper addresses the second research question and applied the ZFEA to analyse and explain SME growth. The study focused on successful SMEs within South Africa and collected data through a multiple case study. A fundamental EA systems perspective, such as the ZFEA that represents the complete enterprise, its underlying parts and elements interactions, is a helpful classification tool to map, understand and explain business-related responses and focus. From the premise that the fundamental EA perspective as represented by the ZFEA could serve as an explanatory theory, it should be possible to understand and explain enterprise transformation. In our multiple case study research using the ZFEA to analyse start-up and growth of 7 successful SMEs using 11 in-depth semi-structured interviews, identifiable focus patterns emerged.

During the start-up phase of the SME, the focus was primarily on the enterprise/instantiation perspective, and also mainly on inventory (what), process (how) and responsibility (who) columns. However, evidence of the strategic executive perspective was detected, as well as the other columns (when, why and where) because all participants were aware of all elements of the enterprise. For some, but not all, cases, distribution instantiation (where) was relevant.

During SME growth and as complexity increased, transformation upwards through the different perspectives was observable with emphases on the executive and business perspectives. During change and growth, there was evidence of focus on the architect, engineer and technician perspectives as complexity increased. From the analysis of the current focus, the complete organisation with all

elements was indicated to be in place, with more cognisance regarding the details and different perspectives needed.

The fact that none of the cases had formal EA practices was an interesting observation echoing the sentiment that EA frameworks, practices and capabilities are not always feasible and are too resource-intensive, especially for SMEs during start-up and growth. This observation supports our premise that EA should play a different role in these cases, namely that of a theory to understand and explain, and this is supported by the fact that when the ZFEA was used to analyse the case SMEs, patterns were detectable. The basic elements of the organisations existed from the start, but during growth, the complexity of these elements increased and interaction between elements became more relevant.

In all cases, there was continuity of resources that drove the focus during start-up and growth, indicating the relevance of a theoretical support framework to enable the replication and monitoring of organisational focus to enable SMEs if the natural ability does not exist. Across all cases, there seemed to be a consistency in focus when comparing the current focus to the focus at the beginning of the organisation and during growth.

*5.3. Generalisability of Findings*

Seddon and Scheepers [127] considered generalisation in IS research, and, although primarily based on quantitative sampling, argue that there is also a need for qualitative studies to discuss boundary conditions. They furthermore proposed a framework with eight possible logical pathways for justifying generalisations in IS research, which primarily present alternatives for statistical quantitative research studies. This study is qualitative, however, there is a similarity to the AB(C) pathway in that our study is based on prior theory and qualitative analysis of the sample from where we develop sample-based knowledge claims. Based on the theory and the detected patterns from multiple case studies, we argue that knowledge claims thought true are also likely to apply in other settings.

With regard to the generalisability of our findings, the establishment of the ZFEA as explanatory theory as done in part 1 of the study extends theory perspectives within the EA domain, and theory is in its nature generalisable. Future research could extend the findings to the architectures of other EA frameworks such as the TOGAF content metamodel [97,128] or the ArchiMate Framework [129]. The initial results of the second part of the study that detected focus patterns are based on the premise of the ZFEA as an explanatory theory. The approach to analyse and understand an enterprise phenomenon is generalisable, but the particular results are country-specific. However, the generalisation and applicability of case study research for application in another setting is well documented, especially if there is consistency across several case studies [114,127,130].

*5.4. Discussion Summary*

In summary, our findings support the argument that the ZFEA could be regarded as an explanatory theory. We then used the ZFEA to analyse successful small- and medium-sized enterprises (SMEs) in order to understand and explain focus during growth. The multiple case study approach used semi-structured interviews developed with the ZFEA as the basis for the analysis. Across participating SMEs, the analysis detected patterns of focus on ZFEA-related organizational elements and concepts during growth transformation, thus supporting the argument that EA may serve as an explanatory theory. If EA theory could assist with understanding and analysing, as well as explaining the outcomes of interventions and transformations, it would be an invaluable basis for organisations within the age of digital transformation.

## 6. Conclusions

In this study, we argue that a fundamental EA that adopts a systems perspective of an enterprise and represents the enterprise as a set of interrelated components (as epitomized by the ZFEA) could be regarded as an explanatory IS theory. Using the classification and structural components of

theory, we analysed the ZFEA and concluded that it meets the requirements of an explanatory IS theory. From this position, the ZFEA could assist with describing, understanding and explaining SME growth and transformation. To further these results, we adopted a multiple case study approach with semi-structured interviews developed from the ZFEA as a data collection method in order to understand SME growth. We identified observable focus patterns during start-up and growth in SMEs that could be used to understand how successful SMEs transform and grow. Our results provide evidence that EA represented through the ZFEA could serve as an explanatory theory for SMEs during start-up, growth and transformation.

**Limitations**: The limitations of the study include the use of only the ZFEA, and although the ZFEA is claimed to be one of the most thorough descriptive EA frameworks, it has not been adopted as extensively as some of the more pragmatic descriptive EA frameworks such as TOGAF (The Open Group Architecture Framework) [97], DODAF (Department of Defence Architecture Framework) [131], ArchiMate [129] or even GERAM (Generalised Enterprise Reference Architecture and Methodology) [109]. Furthermore, the focus pattern identification in our case studies was on SMEs only, which are enterprises with very specific characteristics.

**Future Research**: In order to extend our findings and address limitations, further research could include a similar evaluation of other EA frameworks from the IS theory perspective, as well as applying EA as a theory to understand other organisations and not just SMEs.

Further research will include the development of a theoretical framework to understand and guide enterprise transformations. Our data pointed to the fact that a holistic enterprise perspective is crucial to success and that if this perspective is lacking, failure may be inevitable because growth impact cannot be managed. Such a theoretical support framework could, therefore, also be used to understand and analyse factors that impede enterprise growth. The data hinted to the fact that, during growth, various factors causing internal misalignment could constrain an enterprise. For example, inventory can be a restraining factor during growth as much as an inefficient process or lack of capacity in responsibility.

A theoretical support framework could potentially include a practical framework with an execution methodology for setting up SMEs.

With regards to EA as an explanatory theory, further research could investigate EA as a basis of analysis for understanding established SME growth theories, as well as investigate the fundamental notion of Zachman that there is a meta-model for all enterprises. Adopting EA as theory provides a new perspective for studying EA itself and organisations, as well as organisational growth and transformation.

**Practical implications**: The practical implications are twofold. In the first place, the research supports the strategic value of EA as a management tool. It shows that practitioners that practice EAM should be well versed in all aspects of the business and that EA as a management tool is much more than business–IT alignment. The strategic guidance being provided by an EA practitioner is critical to allow the executive to make the correct alignment decisions both internal and external to the business. The research emphasizes the role of an EA practitioner as being at a strategic level directly relevant for executives. EA as a management tool underscores the need for executives to be versed in the understanding of the business as a system represented through an appropriate framework. The ability to have a reference theory from which to understand, plan and execute central control points in a business is crucial for alignment and business coherence. EA applied in this context provides a strategic basis for planning, alignment, communication and execution of strategic direction. It allows one to model and understand dependencies between business elements during the execution of a strategy and allows one to sequence execution based on dependencies between elements.

Secondly, the research provides a platform for designing work creation interventions by, for instance, government, an organised community or civil groups. The research alludes that it should be possible to replicate specific business designs. Central control regarding key business elements is possible and, if modelled appropriately, can be adjusted across geographical areas and businesses

simultaneously and in an organised fashion. This implies that business support programs can be managed centrally in a practical, cohesive and value adding manner without relinquishing individual business control.

In summary, the findings of our research indicate that EA should be positioned in the strategic business management realm as a crucial tool to understand the system that is being managed by the executives. Positioning ZFEA as an IS explanatory theory provides insight into the role and purpose of the ZFEA in particular (and EA in general), and could assist researchers and practitioners with mediating the challenges experienced by SMEs in particular, and—by extension—enhance sustainability since research indicates that successful SMEs encourage sustainability considerations [5,6,87].

**Author Contributions:** Conceptualization, A.G., A.v.d.M. and P.l.R.; methodology, A.G., A.v.d.M. and P.l.R.; software, P.l.R.; validation, A.G., A.v.d.M. and P.l.R.; formal analysis, P.l.R., A.G.; investigation, A.G., A.v.d.M. and P.l.R.; resources, P.l.R.; data curation, P.l.R.; writing—original draft preparation, A.G.; writing—review and editing, P.l.R., A.v.d.M. and Lorinda Gerber (language editor); visualization, P.l.R.; supervision, A.G. and Alta van der Merwe; project administration, P.l.R. All authors have read and agreed to the published version of the manuscript.

**Funding:** This research received no external funding.

**Conflicts of Interest:** The authors declare no conflict of interest.

## Abbreviations

| | |
|---|---|
| EA | Enterprise Architecture |
| EAM | Enterprise Architecture Management |
| IS | Information Systems |
| IT | Information Technology |
| SMEs | Small- and Medium-Sized Enterprises |
| TOGAF | The Open Group Architecture Framework |
| ZFEA | Zachman Framework for Enterprise Architecture |

## Appendix A.

**Table A1.** Interview Questions and Purpose.

| Question | Purpose |
|---|---|
| What year did the organisation start? | • To provide a guideline for comparison between organisational focus over time and to provide background and context per organisation. <br> • To provide a view on the age of the organisation as an indicator of success |
| How many employees do you currently have? | To confirm that the organisation fit within the parameters for a small- to medium-sized enterprise as defined in this study. |
| Does your current turnover range fall into one of the below categories? <br> • Between R2 million and R64 million. <br> • Between R64 million and R128 million. | To confirm that the organisation fit within the parameters for a small to medium enterprise as defined in this study. |
| What industry would you describe your organisation to be in? | To provide a point of reference for comparing different organisations, and to provide a point of reference to determine whether potential focus patterns emerge across industries. |
| Are you/were you responsible for the success/growth of the organisation? <br> Please give an overview of how. | To determine the applicability of the data source as a driver of organisational success and growth for the organisation. |
| Are you familiar with the concept of EA and holistic systems thinking? <br> Kindly provide your definition of what these concepts entail. | • To enable a comparison of focus between data sources knowledgeable about EA and those potentially not knowledgeable. <br> • To provide a base from which to contextualise the answers, as a difference in understanding may lead to a different description. <br> • To gauge the general awareness of EA and EA concepts in the organisation. <br> • To provide a base from which to analyse answers and help with the analysis of whether EA thinking contributed to the organisation. |

**Table A1.** *Cont.*

| Question | Purpose |
|---|---|
| Please provide an overview and history of how the organisation originated, being specific about what aspects of the organisation you focused on when the organisation started. | • To determine, in as much detail as possible, what elements of the organisation were focused on during start-up. Participants were requested to provide as much detail as possible and to provide substantiating documents if possible. |
| Please list the pertinent growth/inflexion points your organisation went through.·<br>• As the organisation grew, describe the aspects the organisation focused on during growth and subsequent operations.<br>• Why were these aspects important during growth?<br>• What aspects of the organisation were impacted most during periods of growth? Describe how you overcame these impacts. | • To establish focal points during growth periods and, thus, enable comparison between cases.<br>• To confirm the view received in the previous question and to correlate the focus between start-up and growth.<br>• To understand whether there is a pattern of organisational impact during growth. |
| Can you provide an example of the relevant focus areas/building blocks for your organisation currently? Do you deem the above different from the focus areas/building blocks at the start?<br>If so, how did they change and why are they different? | • To gain insight and data regarding the current focus of the organisation to allow for the data to be mapped to the ZFEA.<br>• To gain insight into whether the focus of the organisation changed from the start of the organisation to the present day. This provided a basis for understanding how the organisation adapted.<br>• To provide a relevant current view of focus to compare. |
| Did you have to adapt your organisation to stay relevant in the market over the years?<br>• Is organisational agility important to you?<br>• When making changes to the organisation, what factors do you consider?<br>• Did you consider the influence of the different aspects of the organisation on each other during changes?<br>• What were those aspects you considered? | • To obtain data regarding the process, focus and management of organisational change.<br>• To understand the thinking during change and whether a structured or unstructured process was followed during change.<br>• To obtain data from which to analyse the impact of underlying organisational elements on the organisation during change. |
| Do you use a specific framework or reference when you adapt the organisation?<br>• Please talk me through your thinking process while adapting the organisation.<br>• Do you actively manage/document this or is it a mental consideration only? | To gain insight into whether there is a line of thinking, discipline or framework that is prevalent during change. |
| What would you describe as the generic/other factors that contributed to the organisation's success? | To allow the participants to provide broader insight beyond the scope of the answers. This formed the basis for additional views to ensure that all views could be gathered. |
| Would you say EA contributed to the growth of your organisation, or would you say understanding the impact of the different parts of the organisation on each other contributed to the success of your organisation?<br>If so, please elaborate. | • To establish the applicability of EA for successful SMEs in South Africa.<br>• To provide a basis to establish the relevance of EA, or lack thereof, in successful SMEs. |
| Anything else you want to add that you may feel is relevant? | To allow another chance to provide further insight and remove potential bias towards the applicability of EA. |

## Appendix B. Participants

**Table A2.** Case Studies: Interview Participants and Additional Data Sources.

| Case | Participants | Other Data Sources |
|---|---|---|
| Case 1 was a consulting organisation located in Centurion, South Africa. Two interviews were conducted for this case. Case 1 delivered business change services to numerous clients across banking, mining and other industries. Implementation of change projects to clients was core to their offering. The organisation had a smaller regional office in Cape Town as well. | A: Director and founder<br>B: Director and founder<br><br>Separate interviews to corroborate the versions and the focus. Each interviewee's focus was separately mapped to ensure it aligns. | Own observations, based on site visits, website reviews and market research<br>Strategic documents, planning documents, and relevant material |

**Table A2.** *Cont.*

| Case | Participants | Other Data Sources |
|---|---|---|
| Case 2 was a telecommunications organisation located in Centurion, South Africa. For this case, two interviews were conducted. The organisation's offerings included a range of data and communications products delivered to the consumer market across South Africa | A: Director and founder B: Director and founder<br><br>Separate interviews to corroborate the versions and the focus. Each interviewee's focus was separately mapped to ensure it aligns. | Own observations, based on site visits, website reviews, and market research Strategic documents, planning documents, and relevant material |
| Case 3 was an auditing organisation located in Pretoria, South Africa. For this case, two interviews were conducted. The organisation offered auditing and accounting services to a large variety of customers across different industries, with clients ranging from small to large listed entities | A: Founder of firm B: Managers that were responsible for growth<br><br>Separate interviews to corroborate the versions and the focus. Each interviewee's focus was separately mapped to ensure it aligns. | Own observations, based on site visits, website reviews and market research Strategic documents, planning documents, and relevant material |
| Case 4 was a materials handling organisation located in Johannesburg, South Africa. For this case, one interview was conducted | A: Founder and sole director of firm Ad-hoc check and corroboration of views from discussion with the auditors of the firm (done with consent, of course) | External corroboration of focus events from auditors of the organisation Own observations, based on site visits, website reviews and market research Strategic documents, planning documents, and relevant material |
| Case 5 was a software development and services company that developed, hosted and maintained software solutions for clients. The organisation was located in Somerset West, South Africa. | A: Founder and sole director of the firm With consent did a corroboration test with the first client of the firm to confirm the views presented | External corroboration of focus events from a client of the organisation Own observations, based on site visits, website reviews and market research Strategic documents, planning documents, and relevant material |
| Case 6 was a financial services company that assisted consumers who had problems repaying debt. The organisation was located in Kempton Park, South Africa. One interview was conducted for this case | A: Founder and sole director of the firm With consent corroborated the version with the service provider of the firm responsible for operations from the start | External corroboration of focus events from the service provider of the organisation Own observations, based on site visits, website reviews and market research Strategic documents, planning documents, and relevant material |
| Case 7 was a construction consulting organisation located in Centurion, South Africa. For this case, two interviews were conducted. The organisation offered services in the construction industry. | A: Director and founder cB: Director and founder<br><br>Separate interviews to corroborate the versions and the focus. Each interviewee's focus was separately mapped to ensure it aligns. | Own observations, based on site visits, website reviews, and market research Strategic documents, planning documents, and relevant material |

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
