# Peer review of "Enterprise Architecture as Explanatory Information Systems Theory for Understanding Small- and Medium-Sized Enterprise Growth"

_sustainability, doi:10.3390/su12208517_

Round 1

Reviewer 1 Report

Dear Authors,

After reviewing the manuscript entitled EA as Explanatory Theory in the Age of Digital Transformation, unfortunately, I found significant verbatim text overlap with another paper:

Aurona Gerber, Pierre le Roux, Carike Kearney and Alta van der Merwe. The Zachman Framework for Enterprise Architecture: An Explanatory IS Theory, Published by Springer Nature Switzerland AG 2020, M. Hattingh et al. (Eds.): I3E 2020, LNCS 12066, pp. 383–396, 2020.

Even though it seems that 3 out of 4 co-authors of that paper are the co-authors of the submitted manuscript, also, it is not acceptable, in my opinion, to submit a paper overlapping on that scale with an already published paper.

Dear Authors,

After reviewing the manuscript entitled EA as Explanatory Theory in the Age of Digital Transformation, these are my comments and suggestions:

General comments

1) The topic of the paper is interesting, but I do not see any connection with sustainability or sustainable development. One prerequisite for publication in Sustainability journal is this connection. Please revise the manuscript by adding a context related to sustainability.

2) The structure of the paper is unusual, with too many theoretical sections and a very brief Discussion section.

3) The reference list does not comply with journal style requirements.

4) The title has no sense. First of all, do not use abbreviations in title. Second, Enterprise Architecture as an explanatory theory for what or whom?

Specific comments

1) Abstract and Introduction

  1. a) Explain your abbreviations.
  2. b) I do not understand how the first reference is listed 82. The reference list has to be done according to position of the reference in the manuscript, not in alphabetical order.
  3. c) Do not repeat in in the Introduction parts of the Abstract.
  4. d) This statement has no sense: “We the used the ZFEA to analyse successful SMEs in order to understand and explain through the identification of patterns of common focus given ZFEA organisational elements.”

2) Background

  1. a) Literature review is more appropriate than Background.
  2. b) “The definitions of exactly what EA entails evolved with the discipline, but nowadays most of the accepted definitions include notions such as that EA is the continuous practice of describing the essential elements of a socio‐technical organization, their relationships to each other and to the environment, in order to understand complexity and manage change [43, 44].”. Is this the definition the authors used in the paper?
  3. c) ZFEA is criticized by many and is accepted mainly in the IT architecture community. The authors should explain their choice in using it.
  4. d) There is a mix of theories or alleged theories which limits the understanding of the authors’ arguments. It seems that there is ZFEA but also ZFEA as an explanatory theory, EA but also EA as a theory and on top of this is the IS theory.

3) ZFEA as Explanatory Theory

  1. a) In table 5, ZFEA and theory components of Gregor, the authors seem to provide arguments for ZFEA as NOT an explanatory theory. This is the case for Testable propositions and Prescriptivestatements.

4) Research Method and Data Collection

  1. a) What means a successful SME? What makes it successful?
  2. b) Why were SMEs chosen for the theory development? How were they selected?
  3. c) I totally disagree with this statement: “Although the SME context for this study is country specific, we argue that the results could be generalized because business circumstances within South Africa are challenging due to a shrinking economic base, junk status, crime and ineffective structural support”. These are reasons not to generalize the results.
  4. d) Table 5. What means Schedule 1?
  5. e) I do not understand how someone can develop a theory based on 7 case studies and 11 interviews, except for very specific contexts or industries, which is not the case here.

5) Results and Findings

  1. a) Sub-section 5.1 belongs to the 4th section, not Results and finding because it is a description of the sample.
  2. b) What means start-up for the authors? The initiation phase in the business life cycle? I asked this because in the literature start-up concept has a very clear definition. How was assessed the stage of the lifecycle for each SME investigated? Because what being successful was not explained, this raise the possibility that one of the investigated SMEs to be not successful during start-up but successful during growth and transformation.
  3. c) Overall, as a result of poor conceptualization, the results are not clear.

6) Discussion

This section does not contain one reference while this is all about discussion, perhaps the most important part of any paper: critically discuss the findings by relating to the literature. This is the all point, what the manuscript bring new from a theoretical and practical point of view.

Author Response

See the attachment Reviewer 1

Reviewer 2 Report

- References must be checked again in detail because of missing data (e.g. date of publishing), e.g. Line 740 or 861

- A research question should be derived and formulated so that the reader quickly realizes which research gap is or should be closed by the article

- Some sources in the text are listed as new studies, but on closer inspection these researches are many years old. For example, references 25, 26, 59 and 70. Anyway, there are recent works in this comprehensive subject area (e.g. by researchers called Härting, Lange, Mendling, Recker, Reichstein, Sandkuhl, etc.)

- Altogether the manuscript should be revised linguistically (style and orthography), as some unclear sentences and also punctuation mistakes have crept in: e.g. punctuation mistakes Line 69, 100 - 105, 107, etc. and spaces are missing, e.g. Line 288, 295, 309, etc.

- Line 82: delete "would"

- In some cases, the formulation of the problem and objectives of the work is unclear due to a lack of justification, e.g. "We adopt an IS perspective on theory where IS is the academic discipline concerned with the study of socio-technical, organizational systems that classically includes four components: task, people, structure (or roles), and technology [54]." --> Why to do so?

- Unclear sentence: "The purpose of IS theories is to analyse, 82 predict, explain and/or prescribe and given the stance on theory in IS, an explanatory theory has as 83 main aim explanation and understanding, which are core to building a discipline [23, 25, 59].

- Figure 1: The Zachman Framework for Enterprise Architecture (ZFEA) visually unfavourable offers no added value --> Please, explain it in more detail!

- The methodology is not sufficiently presented and leaves questions open:

o The abstract and the introduction talk about semi-structured interviews within the framework of the methodology, whereas chapter 4.2 Data Collection and Analysis talks about open-ended interview structure --> Where is the mistake?

o Who exactly were the interview partners / test persons? If they were experts, what distinguished the test persons as experts? How were the test persons acquired?

o "The data was collected from 7 different cases, all are successful SMEs operating primarily within South Africa, and 11 interviews were conducted mainly with founders and executives currently still involved in the SMEs." --> Why these companies, why South Africa?

o "Data collected was non-numeric qualitative data, for which there are some guidelines but no firm set of rules in terms of analysis" --> which guidelines?

o "All conducted interviews were recorded and transcribed in order for the text to be analysed." --> How exactly? How was the interview recorded and later transcribed?

o Which analysis tool was used? How did the authors proceed?

General question that might be discussed: Does a start-up or a SME need an EA or an EAM, with special focus on the industry in SA?

Author Response

See the attachment Reviewer 2

Round 2

Reviewer 2 Report

Dear authors

thank you very much for the nice contribution and the extensive reworking. After all the comments and remarks have been taken into account, I believe that the manuscript has not only improved, but can also be accepted in its form.

With best regards

C. Reichstein

Author Response

Sustainability

Dear academic editor,

Please find below our response to the editor’s feedback. We also submit the revised version of our paper as requested.

Kind regards

Aurona, Pierre and Alta

Comment

Line 15. EA emerged. What do you mean, the practice, the concept, the artifacts?.

Response

We found this phrase on line 150 and we revised the manuscript to ensure that we clarify such phrases. “EA as a research field emerged…”

Comment

Line 186. You state that: “EA as a discipline requires theory..” Please give this section a bit more body by incorporating the recent debates concerning the theoretical anchoring of EA in the IS literature. The literature that you use is already a couple of years old, and new discussion arrived. E.g., the resource-based view of the firm or dynamic capabilities, highlighting the role of EA resources and capabilities in that regard, see Shanks (2018) and Van de Wetering, R. (2019)

References:
Shanks, G., Gloet, M., Someh, I. A., Frampton, K., & Tamm, T. (2018). Achieving benefits with enterprise architecture. The Journal of Strategic Information Systems, 27(2), 139-156.
Van de Wetering, R. (2019). Enterprise Architecture Resources, Dynamic Capabilities, and their Pathways to Operational Value. ICIS 2019 Proceedings.

Response

We added a subsection, section 2.3 (L 2.3) that specifically address literature and EA as theory, including how the suggested references support EA as theory.

Comment

Line 289. You argue that: For this article, an EA framework that represents the total set of enterprise elements is needed to enable comparison between SME’s. Is this not a bit weird as you could argue that not every artifact of the framework is even useful in a particular context?

Response

We added the following to clarify this statement (L. 322)

For a study that argues that EA may serve as an explanatory theory, an EA framework that fundamentally represents the total set of enterprise elements is required to enable comparison. Even though certain elements may be less pronounced in certain contexts it does not mean they do not exist. The same way that all elements on the periodic table exists in nature as a total system, the enterprise as a total system has a definitive number of elements, whether explicit represented or not [86]. If we want to explain aspects of businesses such as SME growth in more than one context or for more than one business, we need the ability to compare structural components. It is this ability of an appropriate EA framework that makes it so useful as an explanatory IS theory and management tool.

Zachman himself placed a lot of emphasis on the fact that EA need to represent all aspects of an enterprise and the ZFEA was specifically developed to fulfil this purpose. Zachman coined what he calls “The Enterprise laws of physics” that includes “The First Law of Enterprise Ontological Holism” namely that “Every Cell of the Enterprise Ontology exists. Any Cell or portion of Cell that is not made explicit is implicit which means that you are allowing anyone and everyone to make whatever assumptions they want to make about the contents and structure of that Cell”. The ZFEA is therefore explicitly described as an ontology or theory of a structured set of elements of an enterprise.

Comment

Section 4.2 Data Collection and Analysis
Did you perform an inter-coder procedure?

Response

No, we did not perform an inter-coding procedure independently since in-depth knowledge of the ZFEA is a prerequisite. The researcher responsible for the data collection and analysis did the initial coding and the coding was subsequently verified by other researchers with an in-depth knowledge of the ZFEA both individually and during a discussion session. We added a clarification (L.479)

Comment

Line 881: this section reads like a summary. Please fix this and develop this section to conform to academic standards

Response

We substantially revised both the Discussion Summary as well as the Conclusion sections (we did not track changes for this revision as we moved text around). Discussion was a discussion of the findings, whilst the conclusion summarizes, indicate limitations, further research and implications. We are not clear on what is considered “to conform to academic standards” as our paper conforms to publications within IS literature. We hope our revision is satisfactory.

Comment

Discussion: I miss some practical implications of your findings. You have put a great deal of effort into theoretical development; however, what should an architect, managers etc. do with all this knowledge?

Response

We included a substantial discussion of practical implications in the conclusion (L. 931)
